# Processing and Characterization of β Titanium Alloy Composite Using Power Metallurgy Approach

**DOI:** 10.3390/ma15175800

**Published:** 2022-08-23

**Authors:** Krystian Zyguła, Marek Wojtaszek

**Affiliations:** Faculty of Metals Engineering and Industrial Computer Science, AGH University of Science and Technology, Al. Mickiewicza 30, 30-059 Kraków, Poland

**Keywords:** titanium matrix composites, powder metallurgy, in situ synthesis, elemental powders, microstructure

## Abstract

The β titanium alloy matrix composite was made from a mixture of elemental metal powders, including boron carbide. During the high-temperature sintering process, in situ synthesis took place as a result of the TiB and TiC reinforcing phases formed. The identification of these phases was confirmed by X-ray diffraction and microstructural analyses. The presence of unreacted B_4_C particles and the surrounding reaction layers allowed for the evaluation of diffusion kinetics of alloying elements using SEM and EDS analyses. The direction of diffusion of the alloying elements in the multicomponent titanium alloy and their influence on the in situ synthesis reaction taking place were determined. In addition, the relationship between the microstructural components, strengthening phases, and hardness was also determined. It was shown that in situ reinforcement of titanium alloy produced from a mixture of elemental powders with complex chemical composition is possible under the proposed conditions. Thus, it has been demonstrated that sufficiently high temperature and adequate holding time allows one to understand the kinetics of the synthesis of the strengthening phases, which have been shown to be controlled by the concentrations of alloying elements.

## 1. Introduction

Titanium matrix composites (TMCs) are increasingly being studied because of their high-strength properties, high stiffness, and good strength at elevated temperatures; moreover, there are advantages to using titanium alloys as a matrix, which are low-density and resistant to atmospheric corrosion [1,2]. Taking into account the listed properties, TMCs are being considered as potential structural materials primarily, in the the aerospace industry and cosmonautics [3]. Basically, TMCs can be divided into continuously reinforced TMCs, which can contain silicon carbide (SiC) fibers, and discontinuously reinforced TMCs, which are reinforced with particles [4]. Such particles are mainly B_4_C, graphite, TiB, TiC, TiN, or SiC. The most common method for producing discontinuously reinforced TMCs is powder metallurgy. Titanium master alloy powder or commercially pure titanium powder is mixed with reinforcing particles and then consolidated at high temperatures [5]. The SiC particles added to TMCs most often form an ex-situ reinforcement, which means the particles added to the powder mixture do not react with the matrix during sintering and no new particles are formed. A far more common method used in TMC fabrication is to use in situ reactions during the sintering process, which leads to the formation of new strengthening phases [6].

Due to titanium’s high reactivity to carbon and boron, the in situ formation of additional strengthening phases through the addition of boron carbide (B_4_C) is possible. Basically, the in situ synthesis follows the exothermic Reaction (1) [7]:5Ti + B_4_C→ 4TiB + TiC(1)

The products of the diffusion-controlled synthesis are TiB and TiC interphases. This approach has also been successfully used in the manufacture of composites from other novel materials such as metallic glasses reinforced with high-entropy alloy particles [8]. Their presence in the matrix primarily affects the enhancement of hardness, stiffness, and strength at elevated temperatures. The additional benefits of using in situ reactions in the fabrication of TMCs are the homogeneous distribution of particles of the strengthening phase and the clean interface between the matrix and strengthening phases, as well as flexibility in chemical composition and the proportion of reinforcement addition.

Several thermally activated mechanisms are responsible for the transfer of material during sintering, resulting in an increase in density. During the densification process, phenomena such as volumetric diffusion, diffusion at the grain boundary, surface diffusion, and viscous or plastic flow occur [9]. The aforementioned mechanisms can be activated during the sintering process simultaneously or sequentially. In general, the early stage of sintering begins with the formation of a neck in the contact area between two adjacent particles. The vacancies are then filled by lattice diffusion of atoms from the grain boundary into the neck region. Diffusion processes in general are complex and depend on various factors, such as the shape and size of the particles; the distribution of the alloying elements in the mixture; the microstructure; and the process parameters, which are temperature, atmosphere, and time.

The currently published research results are focused on developing parameters for fabricating TMCs and characterizing their microstructure and basic strength properties. These studies mostly focus on the reaction between commercially pure Ti powder and B_4_C particles. The research results currently presented in the publications are aimed at determining favorable sintering conditions that will allow the composite to achieve high density and produce a large amount of additional reinforcing phase. In works [10,11], the authors showed what effect the addition of a reinforcing phase has on composite density in the spark plasma sintering (SPS) process. Regardless of the amount of B_4_C, a minimum density of 99% was obtained. Tensile strength at room temperature decreased as the amount of B_4_C introduced increased. On the other hand, it increased at elevated temperatures. An increase in the amount of the reinforcing phase increased the microhardness, regardless of the temperature conditions of the test.

Research on the use of β titanium alloys for a matrix is conducted less often. In this research area at most, the use of master powders was attempted. Grützner et al. [12,13] undertook a study to characterize the reaction kinetics of the synthesis of a Ti-5Al-5Mo-5V-3Cr master alloy powder with B_4_C particles. The mixture was also consolidated by the SPS process. The presence of the TiB, TiB_2_, and TiC phases, as well as unreacted B_4_C particles, was confirmed. The microhardness and compressive strength were studied, and the addition of a reinforcing phase increased both of these properties.

Some research groups [14,15,16] have undertaken the use of additive manufacturing methods to produce TMCs. In this case, the reinforcement also results from an in situ synthesis between titanium and B_4_C particles. Prior to the selective laser melting (SLM) process, titanium powder and B_4_C particles are intensively mixed, resulting in the particles of the strengthening-phase coating the surfaces of the titanium powder particles. Such a prepared mixture is used as raw material in the SLM process. It has been shown that by using a such novel method, it is also possible to synthesize the in situ TiB and TiC phases. The produced products have a high relative density and a microstructure typical of additively manufactured materials, where melt pools are clearly visible. The addition of B_4_C particles in SLM samples increases the tensile strength but causes a significant decrease in ductility. In addition, it should be noted that as the content of the strengthening phase increases, the tensile strength gradually decreases.

Despite many recent studies, there are insufficient knowledge and research results on the behavior of a multi-component mixture of elemental powders. Therefore, this paper presents an analysis of the fabrication process of a composite based on β titanium alloy strengthened in situ, through the synthesis reaction of boron carbide with titanium. The initial material was a mixture of elemental powders with a chemical composition corresponding to Ti-5Al-5Mo-5V-3Cr alloy and B_4_C particles. The effect of the mixture preparation conditions on the microstructure after the high-temperature sintering process was discussed. Then, X-ray dispersive spectroscopy was used to identify the phase composition of the obtained material. Using scanning electron microscopy and EDS analysis, the kinetics of elemental diffusion during sintering was described. The hardness of the composite was also compared with that of the unreinforced material, and the effect of nucleating reinforcing phases on it was determined.

## 2. Materials and Methods

Elemental powders of titanium (size < 150 μm, 99.9% purity) aluminum (size < 35 μm, 99.8% purity), molybdenum (size < 35 μm, 99.9% purity), vanadium (size < 150 μm, 99.9% purity), chromium (size < 65 μm, 99.2%), and (as a strengthening phase) boron carbide B_4_C particles (size < 100 μm, 98%) were used in this study. The morphology of the powders is shown in Figure 1. To prepare the mixture, the elemental powders were weighed at a ratio appropriate to the chemical composition of the Ti-5Al-5Mo-5V-3Cr alloy with 2 wt.%. (about 3.58 vol.%) addition of B_4_C. Then, powders were mixed in a ceramic mixing chamber in the presence of 8 mm diameter tungsten carbide balls. The mass ratio of the balls and the mixture was 1:1. The mixing process was carried out for 90 min, with a mixer speed of 55 rpm. Additionally, a reference mixture was prepared without the addition of B_4_C. Then, the mixtures were cold compacted under a pressure of 450 MPa and directly subjected to pressureless sintering at 1250 °C for 4 h in a protective atmosphere of argon. The material was then slowly cooled with the furnace. The density of sintered samples was measured based on the Archimedes method and was 4.03 ± 0.05 g/cm^3^ for composite and 4.15 ± 0.04 g/cm^3^ for the unreinforced Ti-5553 alloy.

X-ray diffraction phase analysis was performed using a Panalytical Empyrean DY 1061 X-ray diffractometer, and a Cu lamp Kα = 1.54 Å, an angular range of 2θ from 20° to 90° with a step of 0.03°, and a scanning frequency of 7 s, at 40kV, 40mA. Samples for microstructural observation were prepared using a standard grinding and polishing procedure and etching with Kroll reagent (2% HF + 6% HNO_3_ + 92% H_2_O). A microstructural analysis was performed on a Leica DM4000M light microscope, Hitachi TM-3000, and FEI Inspect S50 scanning microscopes; both microscopes were equipped with an energy-dispersive spectrometry system (EDS). Hardness measurements were carried out by the Vickers method on a Duramin-40 hardness tester, using an indenter load of 19.62 N and for microhardness tests, 0.25 N. A hardness distribution map was prepared in Surfer 17 software using the Kriging griding method.

## 3. Results and Discussion

### 3.1. Mixture Preparation

Figure 2 shows elemental distribution maps for a mixture of elemental powders. Powders with different particle sizes were used for the study. The use of tungsten carbide balls had the effect of intensifying the mixing process. Larger and harder particles of titanium and alloy powders became finer, which allowed for the homogeneous distribution of powder particles in the mixture volume. It was noted that after the mixing process, the smaller particles filled the empty spaces between the large particles. Such a phenomenon has a positive effect on bulk density and enables the achievement of higher density of the green compact. Compared to the morphology of the initial powders, it was noted that the B_4_C particles were not crushed. The results of the EDS analysis confirmed the mixing effects, previously noted during observations of the powder mixture morphology. The effects induced during mixing with the presence of WC balls, involving the crushing of the particles of the individual powders and the inserting of smaller and softer particles, particularly aluminum and molybdenum, onto the surfaces of the larger particles such as titanium, are evident. Thus, it was confirmed that the proposed method of mixing elemental powders, including the use of tungsten carbide balls, leads to a more uniform mixture in terms of size and results in the acceleration of the diffusion process of alloying elements during sintering and acquisition of better homogeneity regarding the chemical composition of the product. However, it should be mentioned that such a method of mixture preparation is exposed to oxygen contamination, which is an α-phase stabilizer. During the crushing of the elemental powder particles, the oxidation of the exposed inner surfaces of the powder particles could occur, as well as enclosing within the particle volume of the crushed oxide layers originating from their surfaces. The resulting increase in oxygen content in the tested material could effectively inhibit the effect of β-phase stabilization by other alloying elements, such as molybdenum, vanadium, or chromium [17]. In addition, the oxygen content in titanium alloys increases ultimate tensile strength and yield strength but decreases elongation [18].

### 3.2. Phase Identification and Microstructure

The XRD patterns for the reference sample and the composite are shown in Figure 3. For the non-reinforced sample, peaks were identified for two phases: hexagonal Ti-α and body-centered cubic Ti-β. The XRD pattern for the in situ reinforced sample showed the presence of both B_4_C and additional phases that nucleated during sintering: TiB and TiC. As a result of the addition of reinforcing particles to the powder mixture, which has high reactivity to titanium, an exothermic reaction occurred during high-temperature sintering according to Equation (1). Compared to the reference material, additional peaks were observed that originated from either B_4_C, TiB, or TiC. Other peaks from the strengthening phases partially overlapped with peaks from Ti-α or Ti-β, resulting in higher intensity compared to the non-reinforced material. The presence of the B_4_C phase indicates that not all boron carbide particles have reacted with the matrix completely.

The optical microstructures of reference material and in situ reinforced composite are shown in Figure 4. The microstructure of the reference material (Figure 4a) was homogeneous and consisted mainly of needle-like α’ phase precipitates on the β phase matrix. The new α_GB_ phase grains first formed at the boundaries of the primary β-phase grains. Then, as a result of slow cooling, the new α” grains nucleated from the primary β-phase grain boundaries to the interior of the grain-forming α” colonies. The microstructure observations revealed significant porosity for both the titanium alloy and the composite. The pores were mainly closed and spherical, and their size did not exceed 100 μm. The exception was unreacted boron carbide particles, which can be distinguished in Figure 4b; they were surrounded by an approximately 20 ± 3 μm reaction layer and a channel void whose length exceeded 100 μm. The locally low consolidation occurring near the B_4_C particles was mainly due to the high melting point of these particles (2350 °C). The sintering of ceramic particles took place at much higher temperatures, whose range was 1800–2200 °C [19,20]. In the case of a titanium matrix composite, this was not possible as the melting point of the main alloy component would be exceeded. These particles were connected to the matrix material via diffusion necks. For the composite, the α-phase morphology had similar characteristics to the reference material. The inner needles of the α”-phase grains were slightly shorter. Additionally the additional strengthening phases precipitated during sintering were noticeable.

Sintering conditions have a key effect on the microstructure of the resulting product, which has a direct impact on the strength properties. The obtained microstructure of the composite matrix is the result of slow cooling from the β-phase field and phase transformations occurring in the metastable Ti-5553 alloy. In general, the lamellar structure is characterized by high strength and fracture toughness but low ductility [21]. Currently, most of the research in the field of the in situ synthesis of TMCs is carried out with the use of commercially pure titanium (CP-Ti) powder and B_4_C particles. In the works conducted by Sabahi Namini et al. [11,22,23], the effect of B_4_C addition on the microstructure and properties of an in situ CP-Ti matrix composite produced by the SPS process was studied. The material was heated and sintered in a β-phase field, but no cooling details were provided. The resulting microstructure of this composite consisted of massive α-phase laths, indicating a relatively slow cooling rate. However, it should be noted that pure titanium has a relatively low strength. The bending strength of such a composite does not exceed 1100 MPa, and it decreases with the increase in the strengthening phase [11]. Therefore, the use of CP-Ti as a raw material is a rather suitable way to study the kinetics of the reaction occurring between titanium and particles of the reinforcing phase. A controlled microstructure morphology can only be obtained with the addition of alloying elements to the matrix material and increased strength properties. In the works [12,13], Ti-5553 master alloy powder and B_4_C particles were used. Sintering was carried out at a temperature above the α + β → β phase transformation, and cooling was uncontrolled. As a result, the matrix microstructure consisted of equiaxial β-phase grains, indicating relatively rapid cooling. The use of the alloy as a matrix material resulted in improved strength properties, and the bending strength increased up to 1600 MPa.

The SEM microstructures of the as-sintered titanium matrix composite are presented in Figure 5. The SEM observations were carried out to analyze the morphology of in situ precipitated strengthening phases. According to the XRD analysis results, the TiC and TiB phases were formed during the sintering process. The EDS point scan was used for the identification of each phase. Due to the fact that the diffusion process was realized at solid-state, the strengthening phases precipitated in the form of colonies than were uniformly distributed on titanium alloy matrix. 

Obtaining reinforcement in the form of TiB and TiC precipitation networks is possible by using a different in situ composite production approach based on powder metallurgy. Wei et al. [24] used graphite powder and TiB_2_ powder as strengthening-phase additives. Through intensive ball milling, the powder particles of the strengthening phases coated the Ti6Al4V powder particles. Thanks to this procedure, during hot-pressing sintering, graphite and TiB_2_ reacted with titanium from the matrix and formed a network of reinforcements across the boundaries of the original Ti6Al4V powder particles. This approach undoubtedly achieves a homogeneous microstructure, but the use of alloy powders is preferred. When elemental powders are used, the layer of graphite and TiB_2_ formed on the surface of the titanium powder during milling can interfere with the diffusion of other alloying elements in the mixture.

TiB precipitated in form of transgranular whiskers (Figure 5c) or grown and elongated blocks was also enriched with carbon (Figure 5b). According to the previous studies [10], the TiB phase precipitated at first by creating the preferred conditions for TiC-phase nucleation due to the high density of stacking faults. TiC precipitates as a form of the equiaxial plates or elongated lamellae on the primary β grain boundary (Figure 5b). It was also noted that the neighborhood of the nucleated reinforcement phases is also carbon-enriched (Figure 5e). In Figure 5d, the unreacted particle of boron carbide is presented. It can be clearly seen that the B_4_C particle is surrounded by the reaction layer and connected to it by the small diffusion neck. Additionally, the reaction layer is connected to the matrix by a visible diffusion neck. Since there is no liquid phase formation during the sintering process, only solid-state diffusion is involved as a densification mechanism. The diffusion process is slowed down by oxide layers naturally occurring on the surface of powder particles. During heating in a resistance furnace, diffusion is slower due to the progressive heating of the material in the direction from the surface to the interior. Therefore, with pressureless sintering, the holding time must be long enough to achieve adequate homogenization. Oxide layers during heating initially break and later dissolve, and then the diffusion of atoms can occur.

To study the diffusion kinetics of boron and carbon during the sintering process, EDS line-scan measurements were performed across both diffusion necks marked in Figure 5d as I and II. Additionally, the EDS mapping of the unreacted B_4_C particle has been undertaken (Figure 6c). The EDS line scan results correspond to the diffusion neck within the B_4_C particle, and the reaction layer (I) is presented in Figure 6a. Naturally, boron and carbon concentration in the unreacted particle is elevated. The closer it is to the diffusion neck, the boron concentration decreases slightly, and the carbon concentration increases, indicating a more intense diffusion of carbon toward the matrix. This observation is also confirmed by the EDS mapping, where a higher concentration of boron is seen in the center of the unreacted particle, and carbon accumulates at the periphery of the particle, closer to the reaction layer. In the case of aluminum and chromium, the opposite direction of diffusion was observed, from the matrix through the reaction layer to the center of the unreacted particle. Aluminum concentrates in the center of the particle, at the same location as boron. On the other hand, chromium concentration is higher only closer to the reaction layer, which coincides with the site of increased carbon concentration. The concentration of the other elements (Ti, Mo, and V) inside the B_4_C particle is very low. The complexity of the elemental powder mixture chemical composition means that in addition to the expected reaction of titanium with boron carbide, reactions between other alloying elements and boron or carbon may also occur. These will depend on the diffusivity of the individual elements relative to the alloying additives and the conditions of the sintering process, such as temperature and time. The problem of reactions occurring between aluminum and boron was the subject of early research in terms of describing the Al-B system [25,26], and in terms of producing aluminum matrix composites [27,28]. In general, it has been shown that the formation of new interphases in the Al-B system will primarily depend on three factors: chemical composition, temperature, and time. The reaction between aluminum and B_4_C will occur as early as around 700 °C, where mainly AlB_2_ is formed. As the temperature increases, more complex interphases such as Al_3_BC and Al_3_B_48_C_2_ form. It should be noted that a successful reaction between aluminum and boron carbide requires a significant holding time at elevated temperatures (48 h or more). Alamdari et al. [28] studied the reaction between pure boron fibers and aluminum with titanium addition up to 500 ppm. They showed that aluminum diffuses very quickly into boron, leading to its complete dissolution. The addition of titanium effectively inhibits the dissolution of boron fibers by forming a TiB_2_ layer on their surface, while no diffusion of titanium into boron was observed. Similar observations result from the studies presented in this work. Aluminum can easily diffuses inside the B_4_C particle, and the reaction layer around it is composed mainly of titanium and boron. The presence of titanium inside the B_4_C particle is practically equal to zero. The situation is similar to chromium, which diffuses from the matrix to the inside of the B_4_C particle. Chromium can react with both boron and carbon but since the reaction of titanium with boron has a significantly lower Gibbs free energy than the reaction of chromium with boron (at a temperature of 1200 °C: about −750 kJ/mol and −200 kJ/mol, respectively), it will therefore be preferred, and chromium will instead react with carbon [16,29].

In Figure 6b, the EDS line scan results corresponding to the diffusion neck between the reaction layer and the matrix material are presented (line II in Figure 5d). The reaction layer is mainly composed of titanium and up to about 12 wt.% of boron, which indicates that it is a TiB layer. Additionally, the increased concertation of vanadium and molybdenum can be noticed. A similar observation was presented previously in [12], where a similar composited was taken under the investigation, but as the initial material, the Ti-5553 master alloy powder was used. The presence of other alloying elements in the reaction layer may inhibit the diffusion of boron further to the Ti matrix. However, due to the high diffusivity of B and C in α_Ti_ [30], at the beginning of the sintering process, those elements move and concentrate on α_GB_ and form a strengthening phase, which can be observed in Figure 5b. Further diffusion of boron at subsequent sintering stages is hindered due to the presence of the reaction layer and the relatively low diffusion of boron in TiB_2_ [31], which is present in the reaction layer. As the reaction neck is approached, the content of alloying elements (mainly Al, Mo, and Cr) increases, and the concentration of boron decreases. The carbon content in the reaction layer and the matrix remains the same. This is because carbon diffuses much faster into the matrix, while boron is retained in the reaction layer around the unreacted B_4_C particle. This phenomenon is responsible for the formation of much more TiC strengthening phases than TiB. As presented in Figure 6b from a distance of 35 μm, the fluctuations in the concentration of the elements composing the matrix depend on the components of the microstructure. The concentrations of titanium and aluminum are higher in the grains of the α phase (dark gray regions) as the latter element is its strong stabilizer. The remaining elements, namely, Mo, V, and Cr, stabilize the β phase, and their concentrations are higher in the light-gray regions.

### 3.3. Hardness Measurements

The addition of B_4_C particles to the elemental powder mixture and the nucleation of additional strengthening phases during in situ synthesis resulted in an increase in hardness compared to the non-reinforced material from 220 ± 16 HV2 to 287 ± 20 HV2. The relatively low hardness of both materials was mainly due to high porosity. To evaluate the effect of in situ nucleated phases on the material properties, a micro-hardness map was prepared for the selected area (Figure 7). The chosen area was free of pores and had various microstructural components typical for the material under study. The highest hardness was measured for the phase, which was identified as TiB (marked as 1) and was 789 HV0.025. The matrix of the composite mostly consisted of colonies of α” phase lamellas, so the hardness for the remaining area is relatively uniform and oscillates between 380 and 440 HV0.025. The exception is the measurement that was taken closer to the pore and in the area between α_GB_ grains, which is a higher fraction of β phase (marked as 2). In these areas, the hardness locally decreased to a value of about 200 HV0.025. The microstructure strongly affects the hardness of the material, especially for titanium alloys. Depending on the manufacturing method, the heat treatment, or the phase composition, the hardnesses of a material with the same chemical composition will differ significantly from one another [32,33]. The micro-hardness results obtained correspond well with those available in the literature. The β titanium alloys in the state after quenching from the range of the presence of the β phase have a hardness in the range of 280–310 HV. Remodeling of the microstructure as a result of a different type of cooling or heat treatment, and consequently an increase in the α phase fraction, results in an increase in hardness in the 460–500 HV range [34]. Grützner et al. [13], who studied a similar composite but with a higher volume fraction of B_4_C (12.9 vol.%) and made from master alloy powder, obtained similar matrix hardness. However, they did not show hardness values for the in situ nucleating phase. Instead, they reported the hardness of the reaction layer, which they identified as a mixture of TiB and TiC, and it was 13.3 GPa (about 1356 HV), which is higher than the hardness of the in situ nucleating TiB phase shown in this work, probably due to the higher content of the strengthening phase.

## 4. Conclusions

In the presented work, the process of in situ synthesis of titanium matrix composite produced from elemental powders was characterized. The strengthening of the material resulted from a reaction between titanium and B_4_C particles that leads to the nucleation of TiB and TiC phases. The analysis and discussion of the obtained test results lead to the following conclusions:Using properly developed process parameters of powder mixture preparation and the fabrication of the β titanium alloy matrix composite, a material with high homogeneity in terms of chemical composition and microstructure was obtained.XRD analysis and microstructural observations showed the presence of TiB and TiC strengthening phases and unreacted B_4_C particles. TiB whiskers and TiC plates were identified. The incomplete reaction between Ti and B_4_C is most likely due to the disruption of the reaction by additional alloying elements added to the mixture in the form of elemental powders. Most of the nucleating strengthening phases were identified as TiC.The presence of unreacted particles and surrounding reaction layers made it possible to study the kinetics of elemental diffusion during sintering. It was shown that in addition to the diffusion of B and C into the matrix, there is a diffusion of Al and Cr in the opposite direction (into the B_4_C particle). The reaction layer consists mainly of Ti, B, and a small amount of Mo and V, which inhibit further diffusion of B into the matrix. The C content of the matrix is high, indicating that its diffusion is not particularly inhibited by the alloying elements.Hardness measurements showed an increase in hardness resulting from the reinforcement. It was shown that the increase in hardness results primarily from in situ nucleated phases and from a characteristic microstructure consisting of colonies of α” phase lamellas.The study showed that through the in situ reaction during sintering, it is possible to reinforce the β-titanium alloy made from elemental powders and that the TiB and TiC synthesis is controlled by the adequate addition of alloying elements.

## Figures and Tables

**Figure 1 materials-15-05800-f001:**
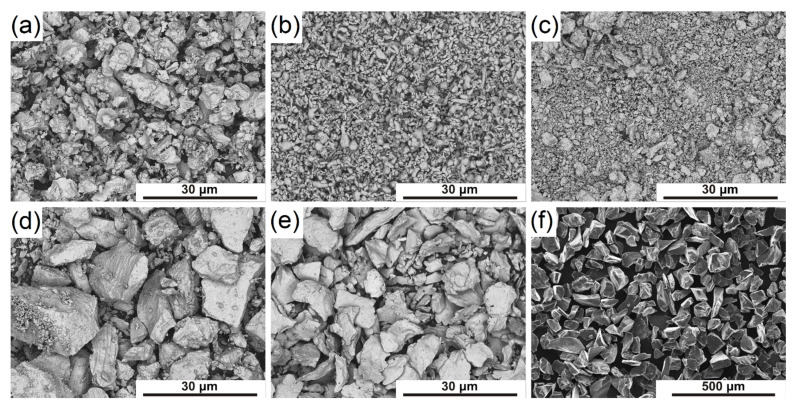
SEM micrographs of elemental powders used to prepare the mixture: (**a**) titanium, (**b**) aluminum, (**c**) molybdenum, (**d**) vanadium, (**e**) chromium, and (**f**) boron carbide.

**Figure 2 materials-15-05800-f002:**
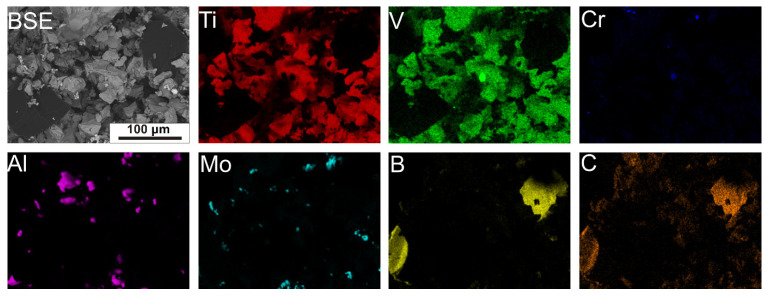
Alloying element distribution maps for a mixture of elemental powders obtained by EDS analysis.

**Figure 3 materials-15-05800-f003:**
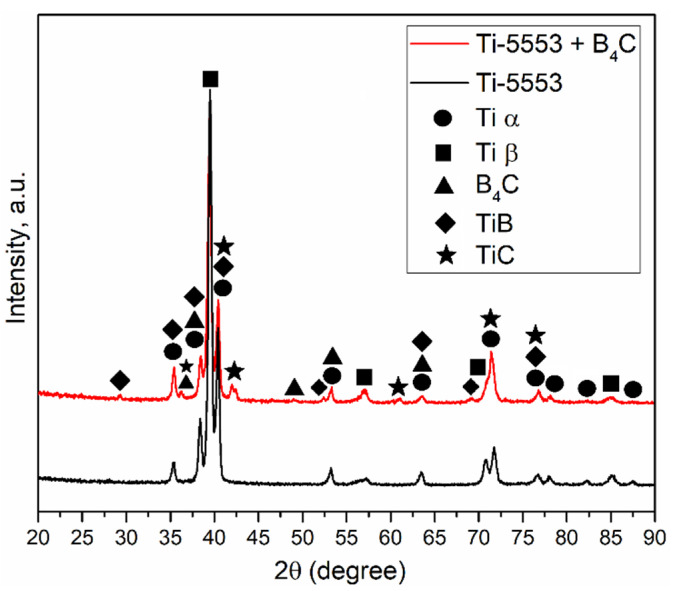
XRD patterns of the Ti-5553 matrix alloy and composite with the addition of B_4_C.

**Figure 4 materials-15-05800-f004:**
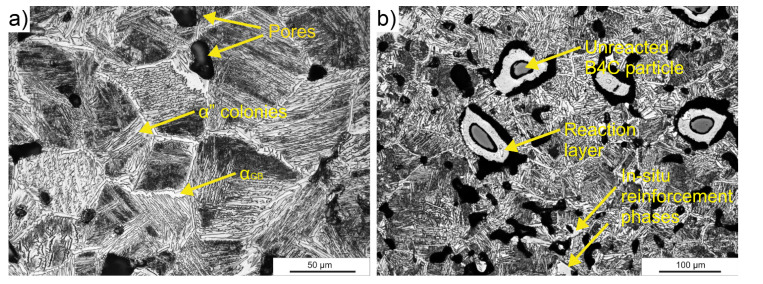
Optical micrographs of as-sintered (**a**) Ti-5553 alloy and (**b**) Ti-5553 + 2% B_4_C composite.

**Figure 5 materials-15-05800-f005:**
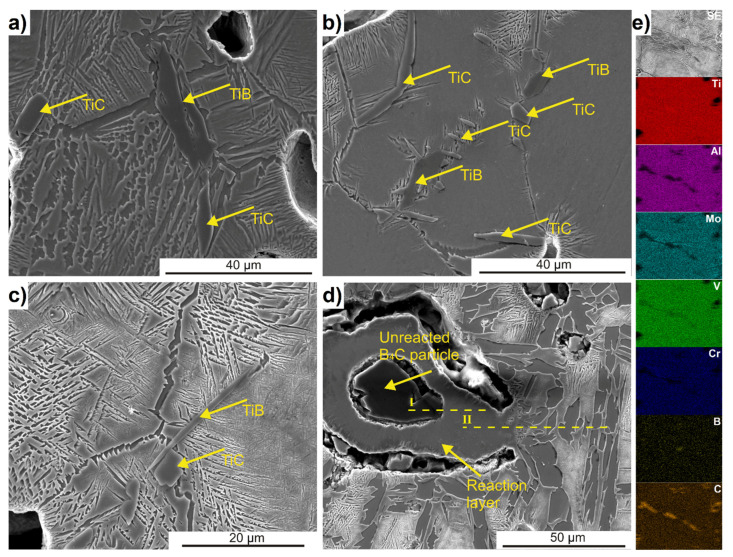
SEM micrographs of Ti-5553 + 2% B_4_C composite with (**a**–**c**) identified different in situ reinforcement phases, (**d**) unreacted B_4_C particle with marked EDS line scan, and (**e**) EDS mapping of TiC phase.

**Figure 6 materials-15-05800-f006:**
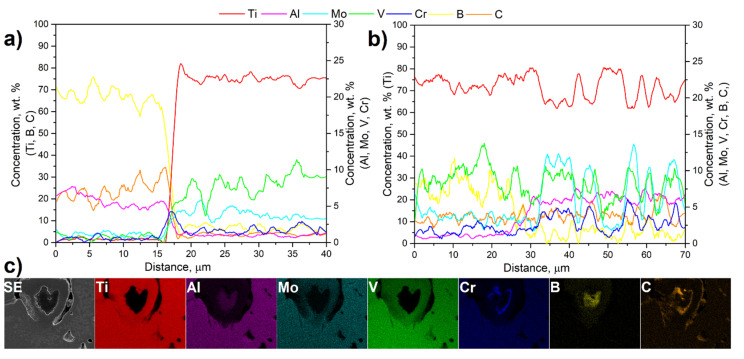
(**a**,**b**) EDS line-scan measurement results of Ti-5553 + 2% B_4_C composite corresponding to Figure 5d; (**c**) EDS mapping of unreacted B_4_C particle.

**Figure 7 materials-15-05800-f007:**
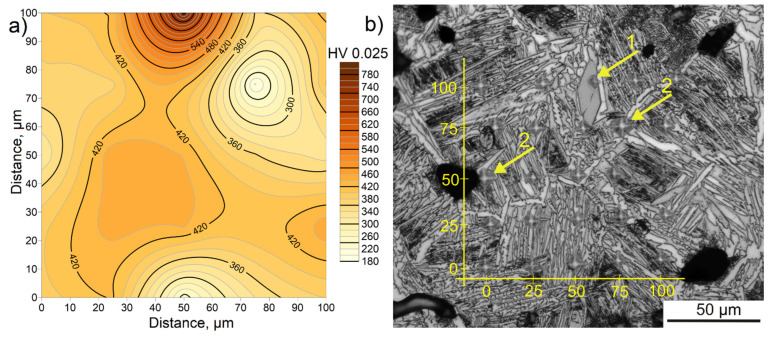
(**a**) Micro-hardness map and (**b**) corresponding microstructure.

## Data Availability

Data sharing is not applicable to this article.

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
