# Peer review of "Processing and Characterization of β Titanium Alloy Composite Using Power Metallurgy Approach"

_materials, 2022, doi:10.3390/ma15175800_

Round 1

Reviewer 1 Report

The authors used a mixture of elemental metal powders and boron carbide to prepare beta titanium alloy matrix composite and experimentally investigated its microstructure and hardness. The manuscript needs several improvements regarding the following points.

1. Since the only reaction between Ti and B4C was conducted in the experiments, Equation (2) should be deleted.

2. The author used the powder metallurgy technique to prepare the alloy. What is the porosity of the titanium alloy matrix composite? How can the author improve the porosity?

3. The microstructure and mechanical property of titanium alloy matrix composite has been investigated by several research groups. Please compare with the literature data and show the advantage of the present work.

4. Grain size has a huge effect on the mechanical property. Thus, electron backscattered diffraction technique should be used to present the corresponding grain size.

Author Response

August 12th, 2022

Dear Reviewer,

I would like to thank you for your review, and the opportunity to resubmit a revised copy of this manuscript. I would also like to take this opportunity to express my thanks for the feedback and helpful comments.

I believe it has resulted in an improved revised manuscript, which you will find uploaded alongside this document. The manuscript has been revised to address your comments, which are appended alongside my responses to this letter.

Below you will find answers addressed to comments and concerns raised in the review.

  1. Since the only reaction between Ti and B4C was conducted in the experiments, Equation (2) should be deleted.

Answer: Thank you for this comment. Equation (2) was mentioned in the manuscript to represent some possible reactions occurring between Ti and B4C during sintering. However, these are not just these two possible reactions, there are more. It was rightly noted that the TiB2 phase was not shown to be present during the material investigation, and the main reaction occurring during sintering is reaction (1). Therefore, equation 2 was removed in the revised version of the manuscript, and the text referring to it was changed.

  1. The author used the powder metallurgy technique to prepare the alloy. What is the porosity of the titanium alloy matrix composite? How can the author improve the porosity?

Answer: The density of the produced composite was 4.03 ± 0.05 g/cm3. In comparison, the density of Ti-5553 alloy (without B4C addition) produced by the same method as the composite was 4.15 ± 0.04 g/cm3. The theoretical density of the composite depends on the degree of reaction occurring between Ti and B4C. When elemental powders are used, the reaction, indicated as (1) in the manuscript, is not the only reaction occurring during sintering. The titanium also reacts with other alloying elements. Therefore, calculating the share of the volume of components obtained, for example, for 100 % conversion of the mixture elements is difficult and may be subject to error. In addition, it should be noted that microstructure observations showed that not all B4C particles introduced into the mixture reacted at 100 %. Nevertheless, it is possible to calculate the theoretical density of the composite for 0 % conversion of the mixture components and this is equal to 4.52 g/cm3 and for 100 % conversion, considering only the reaction between Ti and B4C (in the manuscript reaction (1)) and this is equal to 4.60 g/cm3. In this way, the simplified porosity of the composite can be determined, which is equal to
12.4 %. However, taking into account the given simplifications, the authors are more inclined to give the density of the tested materials, which was measured by the Archimedes method.

Improving the density of material made from metal powders can be realized by using pressure-assisted sintering (e.g., Hot Isostatic Pressing). The second method is to subject the material produced in the proposed method to hot deformation, which will allow the material to be densified to 100 %.

  1. The microstructure and mechanical property of titanium alloy matrix composite has been investigated by several research groups. Please compare with the literature data and show the advantage of the present work.

Answer: Thank you for this suggestion. The additional discussion of the state of the microstructure of produced composite was added to Section 3.2. The manuscript presents the results of the hardness measurements of the manufactured composite and they are compared with the available literature in Section 3.3. No additional strength tests were undertaken as part of this work. Instead, the focus was on analyzing the in-situ synthesis of strengthening phases during sintering, thus demonstrating the feasibility of producing such composites from a multi-component mixture of elemental powders through a relatively simple process of cold compaction and pressureless sintering.

  1. Grain size has a huge effect on the mechanical property. Thus, electron backscattered diffraction technique should be used to present the corresponding grain size.

Answer: The authors agree that grain size is a key factor regarding mechanical properties. Given that strength tests such as tensile or compression were not performed as part of this work, there was no reason to correlate grain size and strength properties in these tests. However, the approximate size of the primary grain of the β phase can be read in Figure 4.

Best regards,

Krystian Zyguła, MEng, PhD,

Department of Metal Forming

Faculty of Metals Engineering and Industrial Computer Science

AGH University of Science and Technology

Al. Mickiewicza 30, 30-059 Krakow, Poland

phone: (+48) 12 617 38 72,

fax: (+48) 12 617 33 39,

Reviewer 2 Report

The present work “Characterization of in-situ synthesis of β titanium alloy compo- 2

site with blended elemental powder metallurgy approach” by Wojtaszek et el, produce results on in-situ synthesis of beta titanium alloy. The authors have presented nice results, however some major revision are encouraged before its publication to the journal. Below given are the comments that may be incoporated while preparing the revised version:

1.       How were the powdeers handled not only during the characterization process but also during the storage period, Please also comment on how the contamination of the powders was prevented?

2.       For the clarity of the readers, details of JCPDS file should be mentioned alongside discussing the XRD results.

3.       In line 126, “Thus….. mixture”, would be good if a reasonbly sound scientific proof is mentioned in the support of this statement.

4.       Is the Figure 4 represent, the images of mixtures only or the sintered product? If sintered, then please also give the sintering cycle, and if not sintered,, then a strong justification is required for the phase formation without sintering.

5.       Introduction part still seems to be less representative of the current study.. would be good if some most recent work on in-situ experimentation is cited and discussed.

6.       Scale bars in Figure 1 are blurred.. would be nice if that is modified in the revised version

Author Response

August 12th, 2022

Dear Reviewer,

I would like to thank you for your review, and the opportunity to resubmit a revised copy of this manuscript. I would also like to take this opportunity to express my thanks for the feedback and helpful comments.

I believe it has resulted in an improved revised manuscript, which you will find uploaded alongside this document. The manuscript has been revised to address your comments, which are appended alongside my responses to this letter.

Below you will find answers addressed to comments and concerns raised in the review.

  1. How were the powdeers handled not only during the characterization process but also during the storage period, Please also comment on how the contamination of the powders was prevented?

Answer: Titanium powder was originally packaged in vacuum bags (1 kg batch). When the bag was opened, a portion of the mixture was immediately prepared from the whole. Alloying element powders and B4C powder were also packaged in vacuum bags by the manufacturer. Once the powder was opened and the required portion weighed out, it was again sealed and stored. Naturally, in order to protect the material from oxidation during sintering, the process was carried out under a protective atmosphere of argon. As part of another study, the oxygen content of as-sintered Ti-5553, prepared under the same conditions, was tested, and its content was 0.38 %.

  1. For the clarity of the readers, details of JCPDS file should be mentioned alongside discussing the XRD results.

Answer: Phase analysis was carried out, based on PDF patterns (Powder Diffraction File) downloaded from the International Center for Diffraction Data (ICDD) database. Along with the microstructure observations, an additional qualitative analysis of the peaks present in the diffractogram obtained from the composite sample was carried out. Based on o materials data provided by Materials Project Data Base it was possible to identify peaks overlapping but coming from different phases.

  1. In line 126, “Thus….. mixture”, would be good if a reasonbly sound scientific proof is mentioned in the support of this statement.

Answer: Thank you for this suggestion. The authors agree that the presented image of the mixture in Figure 2 may not be sufficient to give the conclusion that the mixture was homogeneous. Rather, the authors intended to state that the use of WC balls resulted in the crushing of the particles which allows for better mixing of the mixture components and partial equalization of their sizes (since particles with a wide size range were used). An additional effect is also inserting soft and small particles, like Al or Mo into larger ones such as Ti which will result in accelerated diffusion of alloying elements during sintering and obtaining better homogeneity of the chemical composition of the product. The conclusions of the observations and the obtained results are clarified in the revised version of the manuscript in Section 3.1.

  1. Is the Figure 4 represent, the images of mixtures only or the sintered product? If sintered, then please also give the sintering cycle, and if not sintered,, then a strong justification is required for the phase formation without sintering.

Answer: Figure 4 shows the microstructures of materials after the sintering process, Ti-5553 alloy and Ti-5553 + 2 % B4C composite. Therefore, two mixtures were prepared separately, with and without the addition of B4C particles. Then, they have been subjected to a cold-compaction process, followed by sintering at a temperature of 1250 ºC for 4h in a protective atmosphere of argon.

  1. Introduction part still seems to be less representative of the current study.. would be good if some most recent work on in-situ experimentation is cited and discussed.

Answer: Thank you for this suggestion. Several other recent references have been included and discussed in the Introduction section of the revised manuscript.

  1. Scale bars in Figure 1 are blurred.. would be nice if that is modified in the revised version

Answer: Thank you for this suggestion. The scale bars in Figure 1 have been changed to more visible ones.

Best regards,

Krystian Zyguła, MEng, PhD,

Department of Metal Forming

Faculty of Metals Engineering and Industrial Computer Science

AGH University of Science and Technology

Al. Mickiewicza 30, 30-059 Krakow, Poland

phone: (+48) 12 617 38 72,

fax: (+48) 12 617 33 39

Reviewer 3 Report

This work is interesting, but some improvements should be made before it can be accepted for publication.

1. How does the sintering parameters affect the sintering behavior and the related properties?It may be better to study the mechanical properties of the sintered samples.

2. Introduction: it is better to cite some recent papers about the preparation of metallic composites by powder sintering. E.g., Materials & design 210 (2021): 110108.

3. It is better to discuss the related densification mechanisms during the sintering process.

4. Figure 2: the bar should be given. It seems the secondary phase does not distribute uniformly.

5. For the experimental data, error bars should be given.

6. A common type of information was presented in the abstract section. Please simply the Abstract section by briefly stating the novelty of the study in one or two sentences. Key findings of the study should be included in the Abstract, e.g. the effect of process temperature and dwell time should be clearly described.

7. The Conclusion part is too wordy, It is better to condense it.

Author Response

August 12th, 2022

Dear Reviewer,

I would like to thank you for your review, and the opportunity to resubmit a revised copy of this manuscript. I would also like to take this opportunity to express my thanks for the feedback and helpful comments.

I believe it has resulted in an improved revised manuscript, which you will find uploaded alongside this document. The manuscript has been revised to address your comments, which are appended alongside my responses to this letter.

Below you will find answers addressed to comments and concerns raised in the review.

  1. How does the sintering parameters affect the sintering behavior and the related properties? It may be better to study the mechanical properties of the sintered samples.

Answer: The authors agree that the sintering parameters have a major impact on the mechanical properties of the obtained material. The authors realize that conducting mechanical tests and comparing their results with reference material is very important. However, mechanical studies concerning this material are part of other ongoing research and cannot be shown in detail at the moment.

  1. Introduction: it is better to cite some recent papers about the preparation of metallic composites by powder sintering. E.g., Materials & design 210 (2021): 110108.

Answer: Thank you for the suggestions regarding a very interesting publication, this and several other recent references have been included in the manuscript.

  1. It is better to discuss the related densification mechanisms during the sintering process.

Answer: Thank you for the suggestions. In the revised version of the manuscript, the basic densification mechanisms occurring during the sintering of metal powders are described in the Introduction section. In addition, the results of microstructure observations have been discussed in terms of possible densification mechanisms.

  1. Figure 2: the bar should be given. It seems the secondary phase does not distribute uniformly.

Answer: The scale bar has been added to the BSE image in Figure 2. The image was taken at relatively high magnification to show the effects associated with the use of WC balls, such as crushing and inserting soft and small particles, like Al or Mo into larger ones such as Ti. The conclusions of the observations and the obtained results are clarified in the revised version of the manuscript in Section 3.1.

  1. For the experimental data, error bars should be given.

Answer: In the revised part of the manuscript, the standard deviation was added to the experimental results.

  1. A common type of information was presented in the abstract section. Please simply the Abstract section by briefly stating the novelty of the study in one or two sentences. Key findings of the study should be included in the Abstract, e.g. the effect of process temperature and dwell time should be clearly described.

Answer: Thank you for the suggestions. The Abstract was refined in such a way as to highlight the most important issues raised in the manuscript and present key findings.

  1. The Conclusion part is too wordy, It is better to condense it.

Answer: The Conclusions section has been shortened to make it more informative.

Best regards,

Krystian Zyguła, MEng, PhD,

Department of Metal Forming

Faculty of Metals Engineering and Industrial Computer Science

AGH University of Science and Technology

Al. Mickiewicza 30, 30-059 Krakow, Poland

phone: (+48) 12 617 38 72,

fax: (+48) 12 617 33 39,

email: [email protected]

Round 2

Reviewer 1 Report

The revision can be accepted.

Author Response

August 16th, 2022

Dear Reviewer,

Thank you for accepting the revised version of the manuscript.

Best regards,

Krystian Zyguła, MEng, PhD,

Department of Metal Forming

Faculty of Metals Engineering and Industrial Computer Science

AGH University of Science and Technology

Al. Mickiewicza 30, 30-059 Krakow, Poland

phone: (+48) 12 617 38 72,

fax: (+48) 12 617 33 39

Reviewer 2 Report

The title of research needs to be changed. which I have correct by myself, in addition, I have done some changes in the abstract too. If authors deems necessary they can keep the same or can do more changes. 

Here are the revised title and abstract. 

Processing and Characterization of β titanium alloy composite using power metallurgy approach.

Abstract: The β titanium alloy matrix composite was made from a mixture of elemental metal powders including boron carbide. During the high-temperature sintering process, in-situ synthesis took place as a result of the TiB and TiC reinforcing phases formed. Identification of these phases was confirmed by X-ray diffraction and microstructural analyses. The presence of unreacted B4C particles and the surrounding reaction layers allowed evaluation of diffusion kinetics of alloying elements using SEM and EDS analyses. The direction of diffusion of alloying elements in the multicomponent titanium alloy and their influence on the in-situ synthesis reaction taking place were determined. In addition, the relationship between microstructural components, the strengthening phases, and hardness were also determined. It was shown that in-situ reinforcement of titanium alloy produced from a mixture of elemental powders with complex chemical composition is possible under the proposed conditions. Thus, it has been demonstrated that sufficiently high temperature and adequate holding time allows to understand the kinetics of the synthesis of the strengthening phases, which have been shown to be controlled by the concentrations of alloying elements.

1) Authors can copy paste above title and abstract. 

2) They need some changes in English throughout the manuscript too.

Once it is done, it is good the accept the paper for publication.

Regards,

Author Response

August 16th, 2022

Dear Reviewer,

I would like to thank you for your review of revised manuscript.

Below you will find answers addressed to comments and concerns raised in the review.

The title of research needs to be changed. which I have correct by myself, in addition, I have done some changes in the abstract too. If authors deems necessary they can keep the same or can do more changes.

Here are the revised title and abstract.

Processing and Characterization of β titanium alloy composite using power metallurgy approach.

Abstract: The β titanium alloy matrix composite was made from a mixture of elemental metal powders including boron carbide. During the high-temperature sintering process, in-situ synthesis took place as a result of the TiB and TiC reinforcing phases formed. Identification of these phases was confirmed by X-ray diffraction and microstructural analyses. The presence of unreacted B4C particles and the surrounding reaction layers allowed evaluation of diffusion kinetics of alloying elements using SEM and EDS analyses. The direction of diffusion of alloying elements in the multicomponent titanium alloy and their influence on the in-situ synthesis reaction taking place were determined. In addition, the relationship between microstructural components, the strengthening phases, and hardness were also determined. It was shown that in-situ reinforcement of titanium alloy produced from a mixture of elemental powders with complex chemical composition is possible under the proposed conditions. Thus, it has been demonstrated that sufficiently high temperature and adequate holding time allows to understand the kinetics of the synthesis of the strengthening phases, which have been shown to be controlled by the concentrations of alloying elements.

1) Authors can copy paste above title and abstract.

Answer: Thank you for the proposed title and abstract sent. The Authors agree that the proposed title reflects well the ideas of the research described in the manuscript. According to the Authors, the abstract also accurately describes the most important aspects and key findings of this work. Taking the above into account, the authors have decided to use the proposed title and abstract mentioned above.

2) They need some changes in English throughout the manuscript too.

Answer: The authors have made efforts to improve the quality of the language in the manuscript. The manuscript was proofread, and grammatical errors have been corrected.

Best regards,

Krystian Zyguła, MEng, PhD,

Department of Metal Forming

Faculty of Metals Engineering and Industrial Computer Science

AGH University of Science and Technology

Al. Mickiewicza 30, 30-059 Krakow, Poland

phone: (+48) 12 617 38 72,

fax: (+48) 12 617 33 39,

Reviewer 3 Report

The revision is OK.

Author Response

(The authors gave the same response as above.)
